# The Effect of Iron-Fortified Lentils on Blood and Cognitive Status among Adolescent Girls in Bangladesh

**DOI:** 10.3390/nu15235001

**Published:** 2023-12-02

**Authors:** Amy L. Barnett, Michael J. Wenger, Fakir M. Yunus, Chowdhury Jalal, Diane M. DellaValle

**Affiliations:** 1Psychology and Cellular and Behavioral Neurobiology, The University of Oklahoma, Norman, OK 73019, USA; 2Pharmacy and Nutrition, Saskatoon, The University of Saskatchewan, SK S7N 5B3, Canada; 3Psychology and Neuroscience, Dalhousie University, Halifax, NS B3H 4R2, Canada; 4Nutrition International, Ottawa, ON K2J 5S2, Canada; 5Health and Human Performance, King’s College, Wilkes-Barre, PA 18711, USA; dianedellavalle@kings.edu

**Keywords:** iron fortification, iron deficiency, cognition, memory, adolescence, Bangladesh, community nutrition, international nutrition

## Abstract

Background: Iron deficiency is highly prevalent in South Asia, especially among women and children in Bangladesh. Declines in cognitive performance are among the many functional consequences of iron deficiency. Objective: We tested the hypothesis that, over the course of a 4-month iron fortification trial, cognitive performance would improve, and that improvement would be related to improvements in iron status. Methods: Participants included 359 adolescent girls attending Bangladesh Rural Advancement Committee (BRAC) clubs as a subsample of a larger double-blind, cluster-randomized community trial in which participants were assigned to one of three conditions: a condition in which no lentils were supplied (NL, n = 118, but which had the usual intake of lentils), a control (non-fortified) lentil condition (CL, n = 124), and an iron-fortified lentil condition (FL, n = 117). In the FL and CL conditions, approximately 200 g of cooked lentils were served five days per week for a total of 85 feeding days. In addition to biomarkers of iron status, five cognitive tasks were measured at baseline (BL) and endline (EL): simple reaction time task (SRT), go/no-go task (GNG), attentional network task (ANT), the Sternberg memory search Task (SMS), and a cued recognition task (CRT). Results: Cognitive performance at EL was significantly better for those in the FL relative to the CL and NL conditions, with this being true for at least one variable in each task, except for the GNG. In addition, there were consistent improvements in cognitive performance for those participants whose iron status improved. Although there were overall declines in iron status from BL to EL, the declines were smallest for those in the FL condition, and iron status was significantly better for those in FL condition at EL, relative to those in the CL and NL conditions. Conclusions: the provision of iron-fortified lentils provided a protective effect on iron status in the context of declines in iron status and supported higher levels of cognitive performance for adolescent girls at-risk of developing iron deficiency.

## 1. Introduction

Iron deficiency (ID) and iron deficiency anemia (IDA) are highly prevalent in South Asia [1], with estimated rates for mild to moderate IDA in children in Bangladesh exceeding 55% [2], a level well above global estimates [3]. Rural areas in Bangladesh have higher rates of IDA than do urban areas [4]. Although supplementation has been used as a strategy for addressing ID in countries like Bangladesh, this strategy has faced a number of challenges, including poor compliance [5]. Additionally, contaminated water, poor hygiene, and lack of bioavailability make supplementation a less sustainable option [6].

A more sustainable alternative to supplementation is iron fortification, and there are many benefits of fortifying staple foods such as lentils with iron. For example, fortification improves the availability of iron-rich food sources in locations which previously have had little access. It also increases the bioavailability of iron to be absorbed [7,8]. Using crops that are regionally available and culturally relevant to the target population, as lentil is to Bangladesh, is beneficial when implementing iron fortification. Many studies have used beans, pearl millet, and lentils, depending on the geographic location and cultural food staples of the target population. By using culturally appropriate staple food crops, the residents of an area are more likely to sustainably integrate the iron-fortified crops into their diet. All of this suggests that the provision of iron-fortified lentils in Bangladesh may be an effective way to address the iron needs of female adolescents, as well as others at risk of ID.

The benefits of addressing ID with either fortified or biofortified foods go beyond improvements in systemic iron levels to include improvements in perception, cognition, and brain function. For example, we [9] showed that the consumption of a salt doubly fortified with potassium iodate and microencapsulated ferrous fumarate by women of reproductive age led to improvements in visual perception, attentional control, and memory. Similar improvements have been found using other iron biofortified staples (as reviewed in [10]). In addition, there is evidence that improvements in the behavioral measures of attention and memory are accompanied by improvements in brain function, as measured using electroencephalography [11,12].

In the present study, an analysis of a subsample from a larger trial [13], our primary concern was to determine whether changes in levels of systemic iron due to the consumption of fortified lentils would be accompanied by changes in cognitive performance. This question was motivated by the findings just discussed, i.e., that improvements in iron status by way of consuming fortified or biofortified components of the diet also produced improvements in both behavioral measures of cognitive function and neural measures of brain function. Changes in systemic levels of iron were documented in the larger trial and those results were consistent with studies documenting improvements in iron status using staple components of diet by way of both fortification and biofortification.

## 2. Materials and Methods

### 2.1. Participants

The current study was an analysis of a sub-sample from a larger double-blind, cluster-randomized community trial [13], which can be referred to for additional methodological details. Figure 1 summarizes the flow of participants through the study and how we arrived at our subsample. Participants for the larger study were recruited from four upazilas (sub-districts) in Bangladesh, including Muktagacha, Mymensingh Sadar, Bhaluka, and Gaffargaon in the Mymensingh district. The study was conducted at the Bangladesh Rural Advancement Committee (BRAC) clubs within the four upazilas. This program provides scholarly education, health education, social education, poverty education, etc., for both young women and men regardless of socioeconomic status and education.

Two upazilas (Gaffargaon and Bhaluka) were selected for the recruitment of participants for the cognitive testing. A total of 21 BRAC clubs in these two upazilas were selected to cover all three intervention conditions. The chance to participate in the cognitive testing was offered to all girls and enrollment continued until the desired sample size (>120 participants per treatment condition, assuming 50% attrition) was reached or exceeded. Sample sizes were determined using an online calculator for clustered designs (http://www.sample-size.net/means-sample-sizeclustered/; accessed on 11 August 2018), assuming α = 0.05, 1 − β = 0.90 and an effect size of 0.70 (based on results in [14]). The final sample for the cognitive tests consisted of 359 adolescent girls aged 10–17 years. Participants were generally healthy and were excluded if they were either pregnant or breastfeeding. A total of 118 girls were included in the no supplied lentil (NL) condition, 124 in the non-fortified control lentil (CL) condition, and 117 in the fortified lentil (FL) condition. The menstrual status of the girls was assessed using a yes/no question at both BL and EL, and the proportion of the girls who reported having reached menarche at both time points as a function of age is presented in Appendix A.

### 2.2. Iron-Fortified Lentils

The fortified lentils used in the present study were small red lentils grown in Saskatchewan fortified with a solution of sodium ferric ethylenediaminetetraacetate (NaFeEDTA), with approximately 13–14 mg of iron per 100 g of lentils. An initial feasibility study was conducted to determine the correct amount and preparation procedure of the lentils, regionally called daal, for adolescent girls in Bangladesh. It was determined that 37.5 g (~200 g cooked) of a thick preparation, which provided 6.9 g of iron, or 86% of the recommended dietary allowance for the younger girls (9–13 years old) and 46% for the older girls (14–18 years old) [15,16] would be the most effective way to add the iron-fortified lentils into the diet of Bangladeshi adolescent girls with a high compliance rate during a 4-month-long feeding trial. The lentils served in the FL condition were fortified with 1600 ppm of iron, while the lentils served in the CL condition contained approximately 75–90 ppm. In the FL and CL conditions, 37.5 g of raw lentils (~200 g cooked) were served five days per week for a total of 85 feeding days. All lentils were served with one cup of cooked rice.

### 2.3. Laboratory Measures

At baseline (BL, prior to the start of the trial) and endline (EL, after 85 days), participants provided self-report measures of demographics and completed a food frequency questionnaire (FFQ) and anthropometrics. In addition, a 6 mL sample of venous blood was taken to assess the iron biomarkers. A blood sample was also taken at approximately day 42; those data were not used in the analyses reported here as there were no cognitive measures taken at that time.

Venous blood samples were collected by a trained phlebotomist using lithium heparinized vacutainers following an aseptic procedure and using a disposable syringe and needle. Vacutainers were transported within 12 h of collection to the International Center for Diarrhoeal Disease Research, Bangladesh (ICDDRB), based in Dhaka. Serum samples were separated and stored at a temperature of 2–8 °C until they were analyzed. All BL blood samples were analyzed at the same time using the same procedures. Later, all EL samples were analyzed at the same time (at the end of the trial) using the same procedures. Blood measures included a complete blood count (CBC), which included measures of hemoglobin (Hb), hematocrit (HCT), mean corpuscular hemoglobin (MCH), mean corpuscular hemoglobin concentration (MCHC), mean corpuscular volume (MCV), red blood cell count (RBC), and white blood cell count (WBC). In addition, assays were performed for serum ferritin (sFt), soluble transferrin receptor (sTfR), and C-reactive protein (CRP). Values of sFt were adjusted when there was evidence for inflammation (CRP > 5 and WBC > 11.5) [17,18]. Total body iron (TBI) was calculated from sFt and sTfR values using Cook’s equation [19]:$$\mathrm{TBI}=\frac{\log_{10}\left(\frac{sTfR\times 1000}{sFt}\right)-2.8229}{0.1207}$$

Additional details of the laboratory assays are reported in [13].

### 2.4. Cognitive Measures

A total of five tasks were used to measure cognitive performance, including the simple reaction time task (SRT), go/no-go task (GNG), attentional network task (ANT), Sternberg memory search task (SMS), and cued recognition task (CRT). We have used these tasks in previous field studies on the effects of iron repletion [9,11]. The tasks were computer-based (programmed by MJW) and presented on laptop computers using DMDX [20], which allowed for the precise control of the presentation of stimuli and millisecond accuracy of measured reaction times (RTs). The SRT is a simple measure of RT that requires no decision-making or discrimination. The GNG assesses sustained attention and inhibitory control. The ANT assesses three levels of attention: low-level attentional capture, mid-level spatial selective attention, and high-level attentional control [21]. The SMS assesses the speed with which a person can search their short-duration memory [22]. The CRT is a standard recognition memory paradigm in which the participant is presented with pictures of common, nameable objects to study and then tested with both studied (old) and unstudied (new) items.

### 2.5. Ethics

Ethical approvals were received from the University of Saskatchewan, Canada (Bio#17–177), Marywood University, USA (IRB#1139116–2) and the Bangladesh Medical Research Council (BMRC/NREC/2016–2019/455), per their respective protocols. Informed written consent and assent were obtained from each participant and their respective parents, and a copy of the signed assent and consent form was given to the participants and parents.

### 2.6. Statistical Analyses

Condition differences at BL in the frequency of occurrence of characteristics such as anemia and ID were assessed using χ^2^ tests. Condition differences in the values of the blood and behavioral variables at BL were assessed using a one-way analysis of covariance (ANCOVA) with conditions (FL, CL, NL) as a between-participants factor and age as a covariate, using Tukey’s HSD tests for post hoc comparisons. Condition differences in the values of the EL blood and behavioral variables were assessed using a one-way ANCOVA with condition as a between-participants factor and age and BL value of the variables as covariates, using Tukey’s HSD tests for post hoc comparisons. Tests of the plausibility that changes from BL to EL in the iron markers were responsible for changes in the behavioral variables were assessed by regressing each of the behavioral change variables onto three sets of predictors: (a) Hb, sFt, and sTfR; (b) Hb, log10(sFt), and sTfR; and (c) Hb and TBI. For each set of predictors, a “best” model was determined using stepwise model selection to maximize the variance accounted for (*R^2^*) with the smallest number of predictors. The final model was selected from among the “best” models based on a non-zero estimate of the slope(s) for the predictor(s) and the highest *R^2^* value.

The final analyses assessed differences in the change of the behavioral variables as a function of the extent to which individuals responded to the intervention. Three response conditions were defined using the biological daily coefficients of variation (CV) for Hb, sFt, and TBI reported in [23]: (a) those whose blood variables decreased by more than the biological daily CV; (b) those whose blood variables did not change (either declining or improving) more than the biological daily CV; and (c) those whose blood variables increased by more than the biological daily CV. Differences in the number of individuals in each response category across the three conditions were assessed using χ^2^ tests. Changes in each of the behavioral variables as a function of response status were assessed using one-way analyses of variance (ANOVAs) with response status (decrease, no change, increase) as a between-participants factor and no co-variates and Tukey HSD tests for post hoc comparisons, with this being undertaken separately for Hb, sFt, and TBI response. All analyses were performed using SAS 9.4 for Linux.

Prior to analyses, the distribution of sFt was checked and found to be non-normal using the Kolmogorov–Smirnov test (*D* = 0.085, *p* < 0.010). When the values were log10 transformed, the distribution was still found to be non-normal (*D* = 0.087, *p* < 0.010). Consequently, we applied the Box–Cox transformation [24]
$$y^{\prime }=\frac{y^{\lambda }-1}{\lambda },y\neq 0$$
with λ = 0.5 (as estimated using transformation regression). This transformation also failed to produce a normal distribution (*D* = 0.178, *p* < 0.010). We then checked all our analyses involving sFt and found that there were no qualitative differences between the raw and either of the transformed values. Consequently, we report all results excepting those of the plausibility analyses using the raw values of sFt.

## 3. Results

Baseline characteristics of the current study’s sample for each of the three treatment conditions are presented in Table 1. There were low rates of anemia (Hb < 12 g/dL) and ID (sFt < 15 μg/L, TBI < 0, see [19]), as well as ID with (Hb < 12 and sFt < 15) and without (Hb > 12 and sFt < 15) anemia. There was also limited evidence for inflammation (CRP > 5 and WBC > 11.5) or for stunting (HA Z < −2, BMIA Z < −2). There was no evidence for any Treatment Condition differences for any of these characteristics.

### 3.1. Baseline and Endline Iron Biomarkers

The results of the analyses of the BL iron biomarkers in this subsample of the larger trial are presented in Table 2. There were no significant differences among the Treatment Conditions on any of the iron status variables at BL. Age was a significant covariate for sFt, RBC, and MCV. The results of the analyses of the EL iron biomarkers are presented in Table 3. Comparing the values for sFt and TBI in Table 2; Table 3, it can be seen that EL values were lower than BL values for both variables, while there was little if any change in Hb and sTfR. These decreases and the lack of increases need to be considered in the context of the differences in the iron biomarkers across the three conditions at EL, along with the extent to which changes in these biomarkers were dependent on BL values, an issue to which we return below. There were significant differences among the conditions at EL for all the variables except for WBC and CRP. Values of Hb, sFt, TBI, RBC, and MCHC were higher at EL for the FL condition than those for the CL and NL conditions. Values of sTfR were lower at EL in the FL condition than in the CL or NL conditions. For HCT, even though the main effect of the condition was significant, the Tukey tests failed to find any significant ordering among the conditions at EL. For MCV and MCH, values for the FL and NL conditions at EL were equivalent and were greater than those for the CL condition. 

### 3.2. Baseline and Endline Cognitive Variables

The results of the analyses of the BL cognitive variables are presented in Table 4. While the main effect of the intervention condition was significant for RTs in the SRT and GNG, and for RTs to center cues and RTs with consistent flankers in the ANT, the Tukey post hoc comparisons failed to find any significant orderings among the conditions. There was a significant main effect for the condition of RTs with inconsistent flankers, as well as for the conflict score in the ANT, with RTs for the FL condition being longer than those in the CL and NL conditions. The only significant main effect for the condition in the SMS was for the intercept of the search function for new items, which was lower in the CL conditions than in the FL or NL conditions. Finally, in the CRT, the only main effect for the condition was for the percentage increase in capacity, which was lower in the FL condition than in either the CL or NL conditions. Age was a significant covariate for most of the variables, excepting the alerting score, the orienting score, and the conflict score for the ANT, the intercept and the slope of the search function for new items and the slope of the search function for old items in the SMS, and the percent change in capacity in the CRT.

Results of the analyses of the EL cognitive variables are presented in Table 5. The main effect for the conditions was significant for several variables, including the RTs for the SRT; the RTs for zero cues, RTs for two cues, RTs for center cues, the orienting score, and RTs with both inconsistent and consistent flankers in the ANT; and the intercept and slope of the search function for old items in the SMS. For the SRT, RTs in the FL condition were shorter than those in either the CL or NL conditions. For RTs in the zero-cue condition of the ANT, although the main effect for the condition was significant, the Tukey tests did not find significant orderings. For RTs with two cues in the ANT, those in the FL and NL conditions were shorter than those in the CL condition. For RTs with center cues in the ANT, those in the FL condition were longer than those in either the CL or NL conditions. However, for the orienting score in the ANT, values for the FL condition were higher (better) than those in either the CL or NL conditions. Although the main effect for the conditions was significant for the RTs with inconsistent flankers in the ANT, the Tukey tests failed to find any significant orderings. For RTs with consistent flankers in the ANT, those for the FL condition were shorter than those for either the CL or NL condition. For the intercept for the search function for old items in the SMS, although the main effect of the condition was significant, the Tukey tests did not reveal any significant orderings. However, for the slope of the search function for old items in the SMS, the values for the FL condition were lower (shallower) than those for either the CL or NL conditions. Age was a significant covariate for the majority of the EL variables, except for the intercepts and the slopes of the search functions for both new and old items in the SMS, and the percent change in capacity in the CRT. BL values of the variables were also significant covariates for most of the variables, except for the alerting score in the ANT, the slope of the search functions for both new and old items in the SMS, and the percentage change in capacity in the CRT.

### 3.3. Plausibility Analyses

Table 6 presents the results of the analyses carried out to assess the plausibility of the change in the cognitive variables being due to the change in the blood iron biomarkers. Recall that this was undertaken by regressing the change in the cognitive variables onto sets of predictors drawn from the set of blood iron biomarkers. A best-fitting model was found for most of the cognitive variables, except for RTs for the zero-cue condition in the ANT, the intercept of the search function for new items in the SMS, and the percentage change in capacity in the CRT (not shown in the table). A change in either sFt or log10(sFt) was identified as a significant predictor for change in most of the cognitive variables. A change in sTfR along with a change in either the raw or transformed values of sFt predicted a change in three variables (RT in the SRT, the alerting score in the ANT, and RT with spatial cues in the ANT), while a change in sTfR alone predicted a change in one variable (the slope of the search function for new items in the SMS). Finally, a change in TBI predicted a change in two cognitive variables, RT in the GNG and the orienting score in the ANT.

### 3.4. Change in the Cognitive Variables as a Function of the Response to the Intervention

Table 7 presents the frequencies with which participants in each of the three conditions were identified as showing either a decrease, no change, or an increase in Hb, sFt, and TBI. For all three iron status variables, there were many more participants who either showed a decrease or no change in their iron status than there were individuals whose iron status improved. Considering this group alone, there were many more individuals whose status improved in the FL condition compared to either the CL or NL conditions, and this was true for all three blood iron biomarkers. The frequency of those whose iron status improved in the FL condition was 2–2.5 times that for the NL condition.

An examination of the distributions of BL values of Hb, sFt, and TBI (see Appendix A) suggests that the change in each of these variables (from BL to EL) was negatively related to values at BL. That is, lower BL values were most likely to be related to positive change. Table 8 presents the means and 95% confidence intervals for BL levels of Hb, sFt, and TBI as a function of response status. For all three variables, the lowest baseline values were observed for those who showed improvements in those variables from BL to EL.

The response status with respect to changes in the Hb failed to predict any changes in the cognitive variables (these results are presented in Appendix A). In contrast, the response status with respect to changes in sFt predicted changes in most of the cognitive variables, with these results presented in Table 9. The exceptions were changes in RTs for 0 cues, changes in RTs for center cues, and changes in RTs with incongruent flankers in the ANT. In all other cases, individuals whose iron measures increased showed improved performance from BL to EL, while individuals whose iron measures either did not change or decreased did not show a change in performance or declined in performance from BL to EL. Similar results were obtained for the response status with respect to TBI, and these results are presented in Table 10. Changes in the cognitive variables differed as a function of response status for the majority of the variables, with the exceptions being changes in RTs for 0 cues, changes in RTs for center cues, and changes in RTs with incongruent flankers in the ANT, and changes in RTs for old items in the CRT. For those variables for which changes differed as a function of response status, those whose measures increased improved from BL to EL while those who showed no change or a decrease in iron status either did not change or showed a decline in performance.

## 4. Discussion

The primary objective of the present study was to determine whether the consumption of an iron-fortified lentil would produce positive outcomes with respect to cognitive performance in a sample of adolescent girls in Bangladesh. Our results indicate that this was the case. Participants in the FL condition were generally faster than those in the CL and NL conditions at EL, with this advantage being present in three of the five tasks. The analyses examining the plausibility of attributing change in the cognitive variables to changes in the blood iron biomarkers were mixed in their results. While there were significant relationships found for many of the cognitive variables (the RTs), these relationships were reasonably weak, accounting for only small amounts of the total variance. The reason for this may be found in the analyses performed to examine the relationship between responses to the intervention in the blood variables and change in the cognitive variables. These analyses revealed that, while most participants showed either negative responses or no response to the intervention (a decline or lack of change in Hb, sFt, and TBI), there were still several participants who showed positive responses (increases in Hb, sFt, and TBI), and the likelihood of showing an increase was related to BL levels, with lower BL levels associated with increases from BL to EL. The majority of those having a positive response to the intervention were in the FL condition, and those who showed a positive response, specifically with respect to sFt and TBI, showed greater improvements (reductions in RTs and an increase in capacity in recognition memory) relative to those who either did not respond or showed a negative response. This was true for all the cognitive variables except RTs in the SRT, and RTs in the zero cue and center cue conditions of the ANT.

Age proved to be a significant covariate in the analyses of cognitive performance at both BL and EL. This is likely due to the fact that, across the age range for this sample (10–17 y), there are significant increases in processing speed, measured as decreases in RTs. It is also the case that this age range includes the mean menarcheal age (15.8 y) for girls in Bangladesh, with there being evidence that there are significant variations in RTs across the phases of the menstrual cycle [25].

An unexpected finding from this study was the overall lack of improvement in any of the iron status biomarkers from BL to EL, which contrasts with the results obtained in the larger study. Indeed, there were decreases in sFt and TBI while there was no change in Hb and sTfR. This also contrasts with the results of prior work with fortified and biofortified foods, in which improvements in iron status have been obtained. This outcome should be interpreted with caution, as the larger study involved a much larger sample, with that sample size determined based on detecting improvements in iron status. That is, the present analysis of the subset of participants may be under-powered to detect improvements in iron status. In addition, at least one important difference between the present study and the previous work is that participants at BL in those studies had either clinically defined ID or IDA, while only a minority of the participants in this analysis had clinically defined ID (4%) or IDA (7%) at BL. Further, it should be noted that the decreases in sFt and TBI were the smallest for the FL condition and that, at EL, the levels of Hb, sFt, and TBI were significantly higher than those in the CL or NL conditions, and that the level of sTfR in the FL condition was significantly lower than those in the CL or NL conditions. It should also be acknowledged that there was an inverse relationship between the BL levels of the biomarkers and the amount of change from BL to EL. Finally, it was suggested that participants’ samples from BL and EL should be re-analyzed at the same time in order to confirm that the changes from BL and EL were reliable and were not confounded with any variations due to measurement time. This is because BL and EL samples were analyzed at different time points (although all BL samples were analyzed at the same time, following BL data collection, and all EL samples were analyzed at the same time, following EL data collection), emphasizing that we used well-validated measures. As re-analysis was not feasible, it must be acknowledged as a potential weakness. A final weakness to be acknowledged is that we did not have information on the phase of participants’ menstrual cycles at the time that blood iron levels were measured. It is unclear as to how this might have influenced the observed outcomes. Thus, whatever the reason for the overall lack of change or decline in iron status from BL to EL, the consumption of the fortified lentils can be seen as having a protective effect against declining iron stores over time, which is in agreement with the findings from the larger trial.

Thus, we conclude that the provision of iron-fortified lentils offered protective or beneficial effects with respect to both cognitive performance and iron status. The preliminary work for this study demonstrated that the daily provision of lentils was acceptable to participants, with a high rate of compliance with the feeding regimen. This suggests that the protective benefits of iron-fortified lentils can be obtained and sustained by integrating these lentils into the daily diet.

A strength of the current study was the reliance on the BRAC clubs within the Upazilas, which allowed for broad and consistent access to the study participants. An additional strength was the use of validated cognitive tests that have been used in a set of studies examining the effects of consuming iron-fortified and biofortified foods on cognitive performance. These studies have shown that the addition of dietary sources of iron resulted in improvements in these cognitive measures, which was the case for those in the present study who showed positive responses to the intervention. An additional strength is that we observed lower levels of inflammation in this sample of relatively healthy young women than has been the case in other studies. The unexpected result of the present study is that levels of sFt and TBI decreased from BL to EL and that Hb levels did not change. Possible explanations for this include not controlling for the age of onset of menarche, resulting in the inclusion of young women who may not have been meeting their increased iron requirements, or regression to the mean.

In sum, the present results suggest that the addition of fortified lentils to the low-iron Bangladeshi dietary pattern may prevent a decline in iron status, thus conferring benefits to cognitive performance. These results offer further evidence that the addition of fortified staple foods is an effective alternative to iron supplementation in environments in which dietary insufficiencies persist. 

## Figures and Tables

**Figure 1 nutrients-15-05001-f001:**
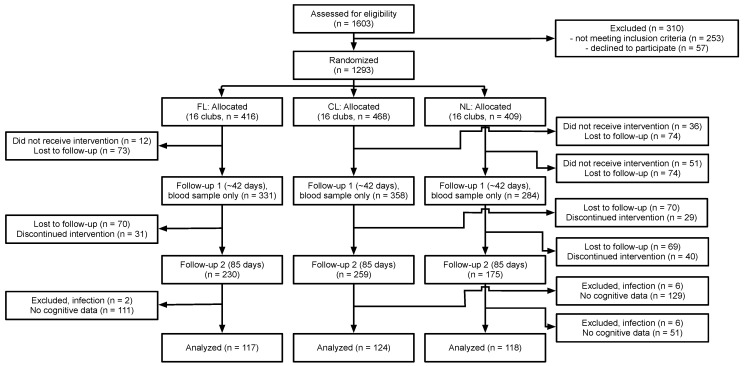
Flow of participants through the study. Note: CL = control lentil, FL = fortified lentil, NL = no lentil.

**Table 1 nutrients-15-05001-t001:** Frequency of baseline characteristics as a function of condition. Entries for age are the median (minimum, maximum), and for all other variables are *N* (%).

Condition					
	FL (n = 117)	CL (n = 124)	NL (n = 118)	χ^2^	*p*
Age (y)	14.0 (10.0, 18.0)	13.0 (10.0, 17.0)	13.5 (10.0, 17.0)		
Hb < 12 g/dL	29 (8)	35 (10)	33 (9)	0.44	0.802
sFt < 15 ng/mL	9 (3)	13 (4)	15 (4)	1.61	0.448
Hb < 12 and sFt < 15	5 (1)	9 (3)	11 (3)	2.34	0.311
Hb > 12 and sFt < 15	4 (1)	4 (1)	4 (1)	0.01	0.996
TBI < 0	4 (1)	8 (2)	5 (1)	1.32	0.516
CRP > 5 and WBC > 11.5	1 (0)	3 (1)	0 (0)	3.32	0.190
HA Z < −2	33 (9)	31 (9)	38 (11)	1.55	0.462
BMIA Z < −2	13 (4)	9 (3)	11 (3)	1.07	0.584

Note: FL = fortified lentil, CL = control lentil, NL = no lentil, Hb = hemoglobin, sFt = serum ferritin, TBI = total body iron, CRP = C-reactive protein, WBC = white blood cell count, HA Z = height-to-age Z-score, BMIA Z = BMI-to-age Z-score.

**Table 2 nutrients-15-05001-t002:** Results of the analyses of the BL iron biomarkers.

Least-Squares Means (SE)									
Variable	Factor	*F*	*df*	*MSE*	*p*	FL	CL	NL	Ordering
Hb, g/dL	Condition	0.60	2	0.71	0.550	12.50 (0.08)	12.38 (0.08)	12.47 (0.08)	FL = CL = NL
	Age	0.25	1		0.618				
sFt, ng/mL	Condition	2.34	2	879.13	0.100	54.35 (2.80)	45.03 (2.72)	50.96 (2.77)	FL = CL = NL
	Age	5.05	1		0.025				
sTfR, μg/mL	Condition	0.09	2	11.72	0.913	3.78 (0.32)	3.92 (0.31)	3.98 (0.32)	FL = CL = NL
	Age	0.35	1		0.553				
TBI, mg/kg	Condition	1.31	2	14.95	0.271	8.00 (0.36)	7.11 (0.35)	7.48 (0.36)	FL = CL = NL
	Age	2.14	1		0.145				
RBC, 10^12^/L	Condition	0.13	2	0.12	0.880	4.62 (0.03)	4.61 (0.03)	4.60 (0.03)	FL = CL = NL
	Age	11.86	1		<0.001				
HCT, %	Condition	2.14	2	8.36	0.119	39.67 (0.27)	38.88 (0.26)	39.43 (0.27)	FL = CL = NL
	Age	0.95	1		0.329				
MCV, fL	Condition	2.32	2	45.50	0.099	257.34 (3.24)	277.39 (3.01)	268.47 (3.26)	FL = CL = NL
	Age	6.09	1		0.014				
MCH, pg	Condition	0.85	2	6.64	0.426	26.91 (0.24)	26.64 (0.23)	27.00 (0.24)	FL = CL = NL
	Age	2.12	1		0.146				
MCHC, %	Condition	1.24	2	1.41	0.292	31.20 (0.11)	31.40 (0.11)	31.35 (0.11)	FL = CL = NL
	Age	2.50	1		0.115				
WBC, 10^9^/L	Condition	2.53	2	5.06	0.081	9.70 (0.21)	10.18 (0.20)	9.59 (0.21)	FL = CL = NL
	Age	0.15	1		0.695				
CRP, mg/L	Condition	0.24	2	5.67	0.788	0.78 (0.22)	0.99 (0.21)	0.91 (0.22)	FL = CL = NL
	Age	0.00	1		0.951				

Note: FL = fortified lentil, CL = control lentil, NL = no lentil, MSE = mean square error, SE = standard error.

**Table 3 nutrients-15-05001-t003:** Results of the analyses of the EL iron biomarkers.

Least-Squares Means (SE)									
Variable	Factor	*F*	*df*	*MSE*	*p*	FL	CL	NL	Ordering
Hb, g/dL	Condition	5.36	2	0.25	0.005	12.32 (0.04)	12.17 (0.05)	12.12 (0.05)	FL > CL = NL
	Age	5.42	1		<0.001				
	Hb, BL	5.42	1		0.021				
sFt, ng/mL	Condition	35.99	2	249.95	<0.001	47.88 (1.57)	37.11 (1.51)	38.13 (1.59)	FL > CL = NL
	Age	1.47	1		0.226				
	sFt, BL	515.82	1		<0.001				
sTfR, μg/mL	Condition	8.23	2	1.35	<0.001	3.67 (0.11)	4.24 (0.11)	3.97 (0.12)	FL < CL = NL
	Age	0.62	1		0.432				
	sTfR, BL	2357.70	1		<0.001				
TBI, mg/kg	Condition	47.47	2	2.49	<0.001	7.37 (0.15)	6.17 (0.15)	6.57 (0.16)	FL > NL > CL
	Age	0.69	1		0.408				
	TBI, BL	1424.86	1		<0.001				
RBC, 10^12^/L	Condition	3.48	2	0.04	0.032	4.52 (0.02)	4.50 (0.02)	4.46 (0.02)	FL >= CL >= NL
	Age	0.68	1		0.411				
	RBC, BL	498.73	1		<0.001				
HCT, %	Condition	3.74	2	3.01	0.025	38.17 (0.17)	38.09 (0.16)	37.70 (0.17)	FL = CL = NL
	Age	5.07	1		0.025				
	HCT, BL	455.50	1		<0.001				
MCV, fL	Condition	9.32	2	4.74	<0.001	84.74 (0.21)	84.98 (0.20)	84.73 (0.22)	CL < FL = NL
	Age	0.38	1		0.536				
	MCV, BL	2567.17	1		<0.001				
MCH, pg	Condition	28.77	2	0.45	<0.001	27.17 (0.07)	26.74 (0.06)	26.88 (0.07)	CL < FL = NL
	Age	0.02	1		0.895				
	MCH, BL	4172.05	1		<0.001				
MCHC, %	Condition	12.84	2	0.37	<0.001	32.04 (0.06)	31.43 (0.06)	31.68 (0.06)	FL > CL = NL
	Age	1.34	1		0.248				
	MCHC, BL	888.63	1		<0.001				
WBC, 10^9^/L	Condition	2.26	2	2.37	0.106	9.10 (0.15)	9.31 (0.14)	9.33 (0.15)	FL = CL = NL
	Age	2.12	1		0.344				
	WBC, BL	194.29	1		<0.001				
CRP, mg/L	Condition	1.90	2	2.68	0.152	0.32 (0.16)	0.76 (0.15)	0.37 (0.16)	FL = CL = NL
	Age	3.61	1		0.058				
	CRP, BL	0.03	1		0.854				

Note: FL = fortified lentil, CL = comparison lentil, NL = no lentil, MSE = mean square error, SE = standard error.

**Table 4 nutrients-15-05001-t004:** Results of the analyses of the BL cognitive variables.

Least-Squares Means (SE)									
Task	Variable	Factor	*F*	*MSE*	*p*	FL	CL	NL	Ordering
SRT	RT, ms	Condition	3.45	1655	0.033	267 (4)	278 (4)	271 (4)	FL = CL = NL
		Age	14.27		<0.001				
GNG	RT, ms	Condition	3.09	3893	0.047	387 (6)	395 (6)	379 (6)	FL = CL = NL
		Age	24.08		<0.001				
ANT	RT 0 cues, ms	Condition	2.11	11982	0.122	609 (5)	611 (5)	605 (5)	FL = CL = NL
		Age	124.08		<0.001				
	RT 2 cues, ms	Condition	2.24	9076	0.107	576 (4)	577 (4)	571 (4)	FL = CL = NL
		Age	137.61		<0.001				
	Alerting score, ms	Condition	0.05	8173	0.951	34 (4)	35 (4)	34 (4)	FL = CL = NL
		Age	1.27		0.261				
	RT center cues, ms	Condition	3.62	11886	0.027	607 (5)	611 (5)	598 (5)	FL = CL = NL
		Age	80.19		<0.001				
	RT spatial cues, ms	Condition	0.95	10241	0.386	568 (5)	564 (5)	561 (5)	FL = CL = NL
		Age	115.04		<0.001				
	Orienting score, ms	Condition	1.44	8433	0.237	39 (4)	47 (4)	37 (4)	FL = CL = NL
		Age	1.41		0.235				
	RT inconsistent flankers, ms	Condition	14.48	21845	<0.001	770 (7)	738 (7)	715 (7)	FL > CL > NL
		Age	99.03		<0.001				
	RT consistent flankers, ms	Condition	4.24	14544	0.015	650 (6)	650 (5)	634 (6)	FL = CL = NL
		Age	99.31		<0.001				
	Conflict score, ms	Condition	9.23	20029	<0.001	120 (7)	89 (6)	81 (7)	FL > CL = NL
		Age	3.58		0.059				
SMS	Intercept new, ms	Condition	4.29	78984	0.014	977 (26)	958 (25)	1060 (26)	CL < FL = NL
		Age	0.31		0.576				
	Slope new, ms/item	Condition	1.36	1482	0.259	47 (4)	42 (3)	50 (4)	FL = CL = NL
		Age	0.38		0.539				
	Intercept old, ms	Condition	0.68	41141	0.505	802 (19)	826 (18)	812 (19)	FL = CL = NL
		Age	4.29		0.039				
	Slope old, ms/item	Condition	0.59	511	0.557	29 (2)	29 (2)	26 (2)	FL = CL = NL
		Age	0.22		0.641				
CRT	RT new, 4 cues, ms	Condition	0.78	21655	0.459	890 (14)	853 (13)	881 (14)	FL = CL = NL
		Age	37.44		<0.001				
	RT old, 4 cues, ms	Condition	0.39	14524	0.676	720 (11)	699 (11)	707 (11)	FL = CL = NL
		Age	21.51		<0.001				
	% change in capacity	Condition	6.73	972	0.001	26 (3)	40 (3)	38 (3)	FL < CL = NL
		Age	0.01		0.916				

Note: FL = fortified lentil, CL = control lentil, NL = no lentil, MSE = mean square error, SE = standard error.

**Table 5 nutrients-15-05001-t005:** Results of the analyses of the EL cognitive variables.

Least-Squares Means (SE)									
Task	Variable	Factor	*F*	*MSE*	*p*	FL	CL	NL	Ordering
SRT	RT, ms	Condition	17.11	980	<0.001	257 (3)	277 (3)	268 (3)	FL < CL = NL
		Age	5.59		0.019				
		RT BL	67.23		<0.001				
GNG	RT, ms	Condition	2.16	1435	0.117	364 (4)	368 (4)	367 (4)	FL = CL = NL
		Age	7.49		0.007				
		RT BL	182.97		<0.001				
ANT	RT 0 cues, ms	Condition	8.52	4386	<0.001	566 (3)	573 (3)	568 (3)	FL = CL = NL
		Age	84.57		<0.001				
		RT BL	482.43		<0.001				
	RT 2 cues, ms	Condition	15.41	3677	<0.001	525 (3)	544 (3)	531 (3)	CL > FL = NL
		Age	14.74		<0.001				
		RT BL	316.24		<0.001				
	Alerting score, ms	Condition	2.84	4041	0.059	46 (3)	31 (3)	41 (3)	FL > CL = NL
		Age	36.41		<0.001				
		Alerting BL	0.23		0.630				
	RT center cues, ms	Condition	4.25	4257	0.015	565 (3)	549 (3)	552 (3)	FL > CL = NL
		Age	64.55		<0.001				
		RT BL	542.65		<0.001				
	RT spatial cues, ms	Condition	1.77	4318	0.171	515 (3)	521 (3)	518 (3)	Fl = CL = NL
		Age	9.84		0.002				
		RT BL	226.46		<0.001				
	Orienting score, ms	Condition	14.15	3534	<0.001	57 (3)	32 (3)	35 (3)	FL > CL = NL
		Age	39.88		<0.001				
		Orienting BL	6.42		0.011				
	RT inconsistent flankers, ms	Condition	4.25	7611	0.014	638 (4)	652 (4)	651 (4)	FL = CL = NL
		Age	45.60		<0.001				
		Rt BL	433.49		<0.001				
	RT consistent flankers, ms	Condition	6.86	5204	0.001	572 (4)	586 (3)	585 (4)	FL < CL = NL
		Age	18.01		<0.001				
		Rt BL	445.94		<0.001				
	Conflict score, ms	Condition	2.19	7479	0.113	74 (4)	65 (4)	66 (4)	FL = CL = NL
		Age	9.39		0.002				
		Conflict BL	73.49		<0.001				
SMS	Intercept new, ms	Condition	0.09	47641	0.910	944 (21)	953 (20)	937 (22)	FL = CL = NL
		Age	0.98		0.322				
		Intercept BL	6.87		0.009				
	Slope new, ms/item	Condition	1.15	2220	0.317	68 (5)	77 (4)	69 (5)	FL = CL = NL
		Age	0.00		0.974				
		Slope BL	2.87		0.091				
	Intercept old, ms	Condition	3.36	23463	0.036	752 (15)	802 (14)	770 (15)	FL = CL = NL
		Age	2.11		0.147				
		Intercept BL	5.95		0.015				
	Slope old, ms/item	Condition	7.81	317	<0.001	32 (2)	42 (2)	38 (2)	FL < CL = NL
		Age	0.93		0.336				
		Slope BL	0.01		0.911				
CRT	RT new, 4 cues, ms	Condition	1.35	15505	0.261	813 (12)	821 (12)	837 (13)	FL = CL = NL
		Age	6.66		0.010				
		RT BL	58.51		<0.001				
	RT old, 4 cues, ms	Condition	2.80	16808	0.063	651 (13)	685 (13)	678 (13)	FL = CL = NL
		Age	7.37		0.007				
		RT BL	6.52		0.011				
	% change in capacity	Condition	0.23	1461	0.793	45 (4)	44 (4)	41 (4)	FL = CL = NL
		Age	0.01		0.936				
		% change BL	1.81		0.180				

Note: FL = fortified lentil, CL = control lentil, NL = no lentil, MSE = mean square error, SE = standard error.

**Table 6 nutrients-15-05001-t006:** Parameters of the regression models used to predict changes in the cognitive variables based on changes in the blood iron biomarkers.

	Change		Predictors: Change Variables					
Task	Variable	Intercept	Hb	sFt	log10(sFt)	sTfR	TBI	*R^2^*
SRT	RT	−8			−12.38	10.49		0.043
GNG	RT	−24					−3.34	0.012
ANT	RT 2 cues	−51		−0.70				0.019
	Alerting score	12			38.06	15.00		0.018
	RT center cues	−52			−12.64			0.003
	RT spatial cues	−55		−0.48		8.62		0.010
	Orienting score	3					3.83	0.004
	RT inconsistent flankers	−102			−27.00			0.006
	RT consistent flankers	−71		−0.67				0.012
	Conflict score	−28		0.37				0.002
SMS	Slope new	23				9.32		0.009
	Intercept old	−57			−80.46			0.026
	Slope old	6			−9.97			0.021
CRT	RT new, 4 cues	−67		−1.03				0.010
	RT old, 4 cues	−49		−1.36				0.020

Note: SRT = simple reaction time task, GNG = go/no-go task, ANT = attentional network task, SMS = Sternberg memory search task, CRT = cued recognition task, RT = reaction time, Hb = hemoglobin, sFt = serum ferritin, sTfR = soluble transferrin receptor, TBI = total body iron.

**Table 7 nutrients-15-05001-t007:** Frequencies, n (row %), with which participants in each of the three conditions were identified as having either a decrease, no change, or an increase in Hb, sFt, and TBI.

Condition	Decrease	No Change	Increase	Row Total
	Hb Response Status			
No lentils	72 (61)	34 (29)	12 (10)	118 (100)
Control lentils	60 (48)	52 (42)	12 (10)	124 (100)
Fortified lentils	48 (41)	48 (41)	21 (18)	117 (100)
Total	180 (50)	134 (37)	45 (13)	359 (100)
	sFt Response Status			
No lentils	91 (77)	12 (10)	15 (13)	118 (100)
Comparison lentils	93 (75)	12 (10)	19 (15)	124 (100)
Fortified lentils	63 (54)	16 (14)	38 (32)	117 (100)
Total	247 (69)	40 (11)	72 (20)	359 (100)
	TBI Response			
No lentils	65 (55)	42 (36)	11 (9)	118 (100)
Control lentils	76 (61)	39 (31)	9 (7)	124 (100)
Fortified lentils	42 (36)	52 (44)	23 (20)	117 (100)
Total	183 (51)	133 (37)	43 (12)	359 (100)

Note: Hb = hemoglobin, sFt = serum ferritin, TBI = total body iron.

**Table 8 nutrients-15-05001-t008:** Means and 95% confidence intervals for BL values of Hb, sFt, and TBI, as a function of response status.

	Response Status		
BL Variable	Decrease	No Change	Increase
Hb (g/dL)	12.68 (12.55, 12.82)	12.33 (12.21, 12.44)	11.94 (11.72, 12.16)
sFt (ng/mL)	54.25 (50.62, 57.89)	51.31 (37.57, 65.05)	35.42 (29.70, 41.13)
TBI	7.50 (6.96, 8.03)	8.64 (8.00, 9.28)	4.17 (3.14, 5.20)

Note: Hb = hemoglobin, sFt = serum ferritin, TBI = total body iron.

**Table 9 nutrients-15-05001-t009:** Changes in the cognitive variables as a function of response status for sFt. Entries for the three categories of response status are the mean change in the variable from BL to EL.

Least Squares Means (SE)								
Task	Change variable	*F*	*MSE*	*p*	Decrease (D)	No Change (N)	Increase (I)	Ordering
SRT	Mean RT (ms)	11.15	1107	<0.001	4 (2)	−10 (6)	−16 (4)	I = N < D
GNG	Mean RT (ms)	51.45	850	<0.001	11 (2)	0 (5)	−30 (4)	I < N = D
ANT	RT 0 cues (ms)	0.64	7876	0.528	−35 (6)	−42 (16)	−49 (11)	I = N = D
	RT 2 cues (ms)	18.99	9086	<0.001	−15 (7)	−32 (17)	96 (12)	I < N = D
	Alerting (ms)	11.06	10,840	<0.001	−20 (7)	−10 (19)	48 (13)	I > N = D
	RT center cues (ms)	2.25	6882	0.107	−43 (6)	−60 (15)	−66 (10)	I = N = D
	RT spatial cues (ms)	25.26	8323	<0.001	9 (6)	−17 (17)	−81 (11)	I < N = D
	Orienting (ms)	16.02	7297	<0.001	−52 (6)	−43 (16)	15 (10)	I > N = D
	RT incongruent flankers (ms)	1.93	11,319	0.147	14 (7)	−25 (19)	16 (13)	I = N = D
	RT congruent flankers (ms)	28.70	12,145	<0.001	5 (8)	5 (20)	−110 (13)	I < N = D
	Conflict (ms)	26.35	15,601	<0.001	9 (9)	−30 (23)	126 (15)	I > N = D
SMS	Intercept, new items (ms)	10.71	109,322	<0.001	66 (22)	99 (59)	−135 (40)	I < N = D
	Slope, new items (ms/item)	31.18	3118	<0.001	32 (4)	12 (10)	−28 (7)	I < N = D
	Intercept, old items (ms)	27.26	40,339	<0.001	58 (13)	−59 (36)	−139 (24)	I < N = D
	Slope, old items (ms/item)	41.93	621	<0.001	20 (2)	14 (4)	−12 (3)	I < N = D
CRT	RT new items (ms)	25.28	24,506	<0.001	18 (11)	−64 (29)	−134 (19)	I < N < D
	RT old items (ms)	15.01	24,700	<0.001	11 (11)	61 (29)	−96 (19)	I < N = D
	Percent change in capacity (%)	56.99	1660	<0.001	−22 (3)	−24 (7)	37 (5)	I > N = D

Note: SRT = simple reaction time task, GNG = go/no-go task, ANT = attentional network task, SMS = Sternberg memory search task, CRT = cued recognition task, RT = reaction time, Neg. = negative, MSE = mean square error.

**Table 10 nutrients-15-05001-t010:** Changes in the cognitive variables as a function of response status for TBI. Entries for the three categories of response status are the mean change in the variable from BL to EL.

Least Squares Means (SE)								
Task	Change Variable	*F*	*MSE*	*p*	Decrease (D)	No Change (N)	Increase (I)	Ordering
SRT	Mean RT (ms)	11.98	1102	<0.001	6 (2)	−6 (3)	−20 (5)	I < N = D
GNG	Mean RT (ms)	19.82	989	<0.001	11 (2)	−1 (3)	−23 (5)	I < N = D
ANT	RT 0 cues (ms)	2.83	7767	0.060	−42 (7)	−26 (8)	−62 (14)	I < N = D
	RT 2 cues (ms)	8.30	9681	<0.001	−12 (8)	−50 (9)	−71 (16)	I = N < D
	Alerting (ms)	9.68	10,931	<0.001	−31 (9)	24 (9)	9 (17)	I = N > D
	RT center cues (ms)	1.47	6917	0.232	−48 (7)	−45 (7)	−71 (13)	I = N = D
	RT spatial cues (ms)	11.02	9036	<0.001	10 (8)	−26 (9)	−63 (15)	I < N = D
	Orienting (ms)	9.55	7584	<0.001	−59 (7)	−18 (8)	−7 (14)	I = N > D
	RT incongruent flankers (ms)	0.25	11,442	0.782	13 (9)	6 (10)	19 (17)	I = N = D
	RT congruent flankers (ms)	11.08	13,433	<0.001	0 (9)	−20 (10)	−98 (19)	I < N = D
	Conflict (ms)	9.90	17,168	<0.001	13 (11)	25 (12)	117 (21)	I > N = D
SMS	Intercept, new items (ms)	4.87	113,160	0.008	55 (27)	39 (30)	−126 (53)	I < N = D
	Slope, new items (ms/item)	14.99	3406	<0.001	31 (5)	13 (5)	−24 (9)	I < N = D
	Intercept, old items (ms)	6.80	45,273	0.001	35 (17)	−1 (19)	−102 (33)	I < N = D
	Slope, old items (ms/item)	3.40	769	0.035	16 (2)	9 (3)	6 (4)	I = N = D
CRT	RT new items (ms)	9.13	26,955	<0.001	9 (14)	−34 (15)	−114 (26)	I < N = D
	RT old items (ms)	1.68	26,879	0.189	4 (14)	−8 (15)	−50 (26)	I = N = D
	Percent change in capacity (%)	13.41	2095	<0.001	−18 (4)	−10 (4)	25 (7)	I > N = D

Note: SRT = simple reaction time task, GNG = go/no-go task, ANT = attentional network task, SMS = Sternberg memory search task, CRT = cued recognition task, RT = reaction time, Neg. = negative, MSE = mean square error.

## Data Availability

Data described in the manuscript, code book, and analytic code will be made available upon reasonable request to the corresponding author.

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
