# Peer review of "The Effect of Iron-Fortified Lentils on Blood and Cognitive Status among Adolescent Girls in Bangladesh"

_nutrients, 2023, doi:10.3390/nu15235001_

Round 1
Reviewer 1 Report
Comments and Suggestions for Authors
General comments:
This is an interesting report from field work implemented in Bangladesh, aimed at improving iron-related nutritional status of adolescent girls, through the iron fortification of lentils, a staple food in that region. The authors evaluated the effect of iron fortification, not only on markers of iron metabolism, but also on cognitive performance. A moderate correlation was found between the improvement of cognitive performance and a positive iron-related response to fortification.
The manuscript is overall well written. The title and abstract are correct and informative and the introduction gives relatively good background for the understanding of the work.
One major drawback of this study is the lack of data on the menarcheal stage of the girls enrolled and the timing of the blood sample collections relative to menstrual cycle. This is particularly relevant in a study involving girls only, with ages between 10 and 17 years old. In general, there is a lack of rigorous description of blood collection and laboratory determinations. Together with the surprising finding of a decline in several iron-related markers through the study period, irrespective of the experimental intervention, this raises concerns as to the reliability of the data.
Main concern:
When interpreting the results of the quantification of iron related parameters, there are two obvious sources of concern:
I- the observation of decreases in virtually all parameters from baseline (BL) to endline (EL), irrespective of the study group;
II- a lack of response to fortification in the majority of the girls fed with fortified lentils.
For me, there are two possible (simultaneous) explanations for these observations:
I- A technical problem in the quantification: no matter how well controlled are the equipments, there might be variations in specific determinations over time. To be sure that there is a decline in, for example, serum ferritin, from BL to EL, all the samples should be quantified at the same time. Was this performed that way? If not, I would recommend that the authors take some samples, including BL and EL samples from the same individual and perform the measurements simultaneously, in order to confirm that the decline is real. In fact, the data on Table 7 (classification of individuals as responders or non-responders) only makes sense if the samples from the same individual were subject to quantification in the same assay, at the same time. If performed this way, it would be interesting to display in the Tables or in graphs (spaghetti plot), the exact variation between BL and EL observed for each girl (and then globally, in each study group).
II- A significant impact of menstrual cycle in serum iron parameters. There are several studies indicating that iron-related serum markers show significant fluctuations along the menstrual cycle in women in reproductive age (reviewed for example in DOI 10.3389/fspor.2022.903937). It is not clear in this study whether the authors had access, for each individual girl, to the information of whether she has reached menarch or not, and if so, in which phase of the menstrual cycle were the samples collected. If these parameters were not taken into account, individual variations may have a significant impact. As the authors mention, the data is different in the “large” cohort, where maybe these type of fluctuations are diluted among the large number of subjects. An interesting way to improve this study would be to get some insight into the hormonal status of the subjects, for example through the measurement of progesterone and estrogen levels in serum samples. The re-analysis of the data in light of these hormonal fluctuations may reveal different patterns of response.
Regarding all the above, the authors should perform the indicated additional assays and/or significantly expand the discussion of the manuscript with the identification of the limitations of the study.
Other specific comments:
1- The authors refer that this is a complementary study to another one involving a much larger number of subjects, referred to as bibliographic reference #12. However, ref. 12 is not correctly identified as a publication and an online search did not retrieve any article with that title. If the work is not published yet, the authors should not refer to it all through the manuscript as if it was. If it is published or accepted for publication, the reference should be corrected.
2- In materials and methods line 80: “13-14mg of iron per 100 Kg of lentils” appears to be a very low value; it is not coherent with reference 18 or with line 114. Please revise.
3- Materials and methods lines 118-131- “Laboratory measures”- More detail should be included:
· How were the samples collected and stored?
· What were the methods/equipments used to perform the quantifications?
· “Values of sFt were adjusted when there was evidence for inflammation (CRP > 5 and WBC > 11.5)”. Specify the ajustments made.
· “Total body iron (TBI) was calculated from sFt and sTfR values using Cook’s equation”- Display the equation please.
4- Table 1- The description of age distribution by the median, minimum and maximal values would be more informative than by the mean and standard deviation.
5- In Table 7, it would be informative to have the percentage of “negative responders”, “non-responders” and “responders” inside each of the study groups (NL, CL and FL).
Minor points:
Line 19 of page 21: There is one word repeated “relative those to those”
Lines 54-59 of page 22: Huge sentence, difficult to follow, should be re-phrased.
The authors’ affiliations must be revised and completed.
References 12, 18 and 27 are not correct or complete.
Author Response
Nutrients ms 2695479: replies to Reviewer 1
We wish to thank Reviewer 1 for the thoughtful and helpful questions and suggestions provided. The following are our specific replies.
- “To be sure that there is a decline in, for example, serum ferritin, from BL to EL, all the samples should be quantified at the same time. Was this performed that way?”
Reply: Each round of blood sample collection, including baseline (BL) and endline (EL), spanned approximately 30 days and encompassed 48 clubs (16 IF, 16 N-IF, and 16 Control). Note that we adhered to a sequential follow-up approach for EL, meaning that if the BL Day-1 blood sample was collected from club-1, we initiated EL Day-1 data collection at the same club-1. This approach allowed us to maintain an approximately equal time gap between blood sample collections timepoints (BL vs EL). The lab that processed our samples stored our daily collected samples in a nitrogen tank at -95 degrees until BL was complete and then conducted tests for all the BL blood samples simultaneously. The same procedure was followed for the endline EL samples. - “… it would be interesting to display in the Tables or in graphs (spaghetti plot), the exact variation between BL and EL observed for each girl …”
Reply: We generated a set of spaghetti plots as suggested and found the results to be unreadable. Instead, we have plotted the change from BL to EL for Hb, sFt, and TBI as a function the BL level for each variable. These plots are presented in Supplementary Figure 2 and show that the lower BL levels of each variable were related to a higher likelihood of obtaining improvement. In addition, we have added a new Table 8 where we show that the lowest baseline values were observed for those who showed improvements in those variables from BL to EL. - “It is not clear in this study whether the authors had access, for each individual girl, to the information of whether she has reached menarche or not, and if so, in which phase of the menstrual cycle were the samples collected.”
Reply: We had a binary response (yes/no) to the question of whether each girl had reached menarche at BL and then at EL. We have added a statement to this effect in the “Participants” subsection of the Methods. We have also added a plot (Supplementary Figure 1) showing the proportion of the participants who reported having reached menarche at BL and EL as a function of age. We did not have any information regarding phase of the menstrual cycle. - “… ref. 12 is not correctly identified as a publication …”
Reply: All references to that document have been removed. - “… 13-14mg of iron per 100 Kg of lentils” appears to be a very low value …”
Reply: This was a typographical error, and has been corrected to read “per 100 g” - Additional details on laboratory measures were requested and have been added.
- “… Cook’s equation”- Display the equation please”
Reply: The equation has been added, as requested. - “The description of age distribution by the median, minimum and maximal values would be more informative …”
Reply: This has been changed as requested. - “In Table 7, it would be informative to have the percentage of …”
Reply: These values were already included. - “Line 19 of page 21: There is one word repeated …”
Reply: This has been corrected. - “Lines 54-59 of page 22: Huge sentence …”
Reply: We were unable to locate the sentence being referred to. - “The authors’ affiliations must be revised and completed.”
Reply: It was not clear to us as to what was missing. - “References 12, 18 and 27 are not correct or complete.”
Reply: Reference 12 was removed, and references 18 and 27 were corrected.

Reviewer 2 Report
Comments and Suggestions for Authors
In this manuscript, the authors present the results of a study exploring whether changes of iron parameters due to the consumption of iron- fortified lentils has an impact on the cognitive performance of subsample of adolescent girls who were included in a previous community-based, double-blind cluster-randomized controlled trial in Bangladesh.
The topic is very relevant, the study is well designed and conducted, and the results are very interesting.
There are some points that need clarification, but overall only minor issues. The reviewer would also suggest some changes in order to improve legibility and an easier comprehension of the data.
Introduction:
- Question: did the authors address/discuss the possible increased bacterial infectious risk related to iron–fortification in their previous experiences? Were there any concerns about this?
Material and Methods:
The structure of this section is somehow confusing. Suggestion for the sub-sections:
· 1 Participants;
· 2 Iron fortified lentils
· 3 Laboratory measures etc…
· “Study design” is not necessary as it is. The first sentence (Line 112-113) belongs actually to sub-section 1; the second part (lines 114-117) belong to sub-section 2
- The authors may consider describing the study design briefly referring for more details to the larger trial, and explaining how the population subsample was selected for the present evaluation etc. Some of this information is already provided in the Introduction.
- “Iron fortified lentils”, line 80: it would be very useful to know which was the iron amount in the 200 g cooked lentils or in the 37.5 g thick preparation, respectively. This data is interesting because it would corroborate the usefulness of the intake of low-dose iron (possibly < 50-60 mg?), an approach that is increasingly adopted for oral iron substitution (especially in children), and that results in a weaker stimulation of hepcidin and in a higher iron absorption in the duodenum
- The timeline of the intervention and its duration are not clear: “…85 feeding days”? 42 days? Other? How were BL and EL defined? Please, add this information also in the abstract.
- “Lab measures” Line 122: why was a blood sample drawn at day 42? Probably this was part of the design of the larger trial. Why were the results not presented here?
- Iron parameters: was a measurement of transferrin saturation also considered? If not, why?
- “Cognitive measures”: this section is too long and rich in specific details (e.g. ANT). The suggestion is to describe shortly the scope and the structure of each test (e.g. number of tasks) and to provide supplemental information with full details (scores etc).
- “Statistical analysis”: same as above.
Results:
- In general, the suggestion is to try and simplify the text and the Tables in order to highlight the key findings. Also, the terms “response” or “responder” (and similar) are better used for the results of the cognitive tests, but not for the changes of laboratory data (e.g. Hb or sFt).
- Table 1: It would be helpful for the readers to have a comment on the variable “TBI < 0” (which means essentially iron deprived tissues) referring to the publication of Cook et al.
- Table 2 and Table 3: These are very detailed Tables. For an easier reading, a selection of the key results of both BL and EL could be presented in one single table, allowing an immediate comparison.The full data can be provided in a supplemental sheet.
- Table 4 and Table 5: same as above
- “Plausibility analysis” and Table 6: also this part could be simplified
- Table 7: (“Negative responders”? The table head is possibly incorrectly formatted). The suggestion is to use “decrease”, “no change” and “increase” instead of “negative responders”, “non-responders” etc. It would be more understandable because data refer to laboratory results.
- Table 8: Maybe convert it into a graph?
Discussion:
The discussion is well structured and addresses all relevant points. Minor remarks:
- Page 21 Lines 13-20: again, changes of laboratory parameters should be better described as increases or decreases etc, and maybe reserve “response”/”no response” to the description of the cognitive tests.
- Page 22, lines 61-64: The decrease of sFt and TBI is not a weakness of the study, but a result.
References:
- Ref. 12 and 13: it is not clear whether the results of the trial were published (no information for Ref. 13)
Author Response
Nutrients ms 2695479: replies to Reviewer 2
We wish to thank Reviewer 2 for the thoughtful and helpful questions and suggestions provided. The following are our specific replies.
- “The structure of this section is somehow confusing.”
Reply: The Methods have been reorganized per the suggestions. - “Study design” is not necessary as it is.”
Reply: Deleted and reorganized as suggested. - “… explaining how the population subsample was selected for the present evaluation …”
Reply: This is now provided in the first sentence of the Participants subsection of the Methods. - “… it would be very useful to know which was the iron amount in the 200 g cooked lentils or in the 37.5g thick preparation …”
Reply: This information has been added. - “The timeline of the intervention and its duration are not clear …”
Reply: The duration of the trial was 85 days, and this is now specified in the Materials and methods and had already been included in the Abstract. - “… why was a blood sample drawn at day 42? Probably this was part of the design of the larger trial. Why were the results not presented here?”
Reply: Yes, this was for the larger trial. We do not report those results as no cognitive measures were taken at this time point, and the focus of this report is on the cognitive outcomes. - “… was a measurement of transferrin saturation also considered? If not, why?”
Reply: In our judgment, transferrin saturation is not among the best indicators of change in iron stores, and TSAT is inferior to the calculation of TBI (from sFt and sTfR), especially in a population with low levels of iron deficiency anemia. Calculation of TBI from sF and sTfR covers the full range of iron status.
See https://www.ncbi.nlm.nih.gov/pmc/articles/PMC5701713/ - ““Cognitive measures”: this section is too long and rich in specific details …”
Reply: The majority of the methodological details have been moved the supplementary information. - ““Statistical analysis”: same as above.”
Reply: We have retained all the detail as, in our judgment, all of these details are needed for a reader to adequately judge the appropriateness and sufficiency of the approach. - “… the terms “response” or “responder” (and similar) are better used for the results of the cognitive tests, but not for the changes of laboratory data …”
Reply: We have changed the terminology to refer to those whose values decreased, showed no change, or increased, and we have made these throughout. - “… referring to the publication of Cook et al. …”
Reply: We have added the necessary citation to the discussion of the Table. - “Table 2 and Table 3: These are very detailed Tables …”
Reply: We have left these Tables unchanged as we believe that there will be readers who will be interested in all of the details. - ““Plausibility analysis” and Table 6”
Reply: The text has been shortened, and non-significant results have been removed from the table. - “The suggestion is to use “decrease”, “no change” and “increase” …”
Reply: These changes have been made as suggested in the text, in Tables 7-9, and in Supplementary Table 1. - “Table 8: Maybe convert it into a graph?”
Reply: We were unable to create a comprehensible graphic and so have maintained this as a table. - “The decrease of sFt and TBI is not a weakness of the study, but a result.”
Reply: This has been changed as requested. - Ref 12 has been removed.

Round 2
Reviewer 1 Report
Comments and Suggestions for Authors
I acknowledge the efforts put by the authors in the revision of the manuscript. This second version includes some important improvements. However, the reply of the authors to my comments was far from complete.
Regarding the main concerns I expressed in the first review
“To be sure that there is a decline in, for example, serum ferritin, from BL to EL, all the samples should be quantified at the same time. Was this performed that way?” The answer of the authors was “no”, as they explained:
Reply: Each round of blood sample collection, including baseline (BL) and endline (EL), spanned approximately 30 days and encompassed 48 clubs (16 IF, 16 N-IF, and 16 Control). Note that we adhered to a sequential follow-up approach for EL, meaning that if the BL Day-1 blood sample was collected from club-1, we initiated EL Day-1 data collection at the same club-1. This approach allowed us to maintain an approximately equal time gap between blood sample collections timepoints (BL vs EL). The lab that processed our samples stored our daily collected samples in a nitrogen tank at -95 degrees until BL was complete and then conducted tests for all the BL blood samples simultaneously. The same procedure was followed for the endline EL samples.
Consequently, I would expect that the authors followed my next comment: “If not, I would recommend that the authors take some samples, including BL and EL samples from the same individual and perform the measurements simultaneously, in order to confirm that the decline is real.” The authors did not answer this part of my comment.
I also added “Regarding all the above, the authors should perform the indicated additional assays and/or significantly expand the discussion of the manuscript with the identification of the limitations of the study.” The authors also ignored this comment.
I do believe that discussion the fragilities of the study will not diminish its interest but will significantly improve the capacity of the reader to interpret the findings, so it will make the publication more useful.
The lack of response to some of my questions may have been due to some misunderstandings, so I will try to clarify them:
(My previous comment #5) In Table 7, the percentages presented inside parenthesis are the percentages relative to the total number of participants. I defend that the percentage inside each study group should be indicated (x% of girls that had FL increased their iron status; Y% of the girls that received FL had decreased iron values…). This means that the total of the percentages in each line should be 100.
Previous comment: Lines 54-59 of page 22 is a huge sentence, difficult to follow, should be re-phrased. This is now on line 56 onwards of page 22: “An additional strength was the use of a set of validated cognitive tests that have been used in a set of studies examining the effects of consuming iron fortified and biofortified foods on cognitive performance, which have shown that the addition of these dietary sources of iron resulted in improvements in these cognitive measures, which was the case for those in the present study who showed positive responses to the intervention.”
The authors’ affiliations must be revised and completed. The only thing I can read in the authors’ affiliations is a name, presumably, of an institution. At least the name of the country (if not a city, an address, a department) should also be included. For example, I found online over 20 institutions in different countries, with the designation “King’s College”.
Additionally, in Table 1, there are no values presented as “mean (standard error)”; that expression should be excluded from the title/legend.
Author Response
Nutrients ms 2695479: replies to Reviewer 1 (revision 1)
We wish to thank Reviewer 1 for the thoughtful and helpful clarifications and suggestions provided on our initial revision. The following are our specific replies.
- “ … I would expect that the authors followed my next comment: “If not, I would recommend that the authors take some samples … The authors did not answer this part of my comment. … I also added “Regarding all the above, the authors should perform the indicated additional assays … The authors also ignored this comment.”
Reply: We sincerely apologize as we did not intend to give the impression that we ignored these comments. Rather, we believed that we had sufficiently addressed the concerns in our reply. We have, as suggested, acknowledged the lack of the re-analysis as a potential weakness in the Discussion: “Finally, it was suggested that participants’ samples from BL and EL should be re-analyzed in order to confirm that the changes from BL and EL were reliable and were not confounded with any variations due to measurement method. As this was not feasible, it must be acknowledged as a potential weakness.” - “In Table 7 … the total of the percentages in each line should be 100.”
Reply: This has been changed as requested. - “Lines 54-59 of page 22 is a huge sentence, difficult to follow, should be re-phrased. This is now on line 56 onwards of page 22 …”
Reply: This has now been broken into two sentences. - “The authors’ affiliations must be revised and completed.”
Reply: City, state (province), and country have now been added for each author. - “Table 1, there are no values presented as “mean (standard error)”; that expression should be excluded from the title/legend.”
Reply: This has been changed as requested.

Reviewer 2 Report
Comments and Suggestions for Authors
Dear Authors, in the reviewer's opinion, the manuscript has improved after further elaboration. The decision of not accepting some suggestions (e.g. Tables) is fine. Well done.
Author Response
We are pleased that the reviewer finds the revision to be acceptable and thank the reviewer again for the thoughtful suggestions.